# The CryoEM structure of the *Saccharomyces cerevisiae* ribosome maturation factor Rea1

Piotr Sosnowski[1,2,3,4†], Linas Urnavicius[5†‡], Andreas Boland[5§], Robert Fagiewicz[1,2,3,4], Johan Busselez[1,2,3,4], Gabor Papai[1,2,3,4], Helgo Schmidt[1,2,3,4]*

[1]Institut de Génétique et de Biologie Moléculaire et Cellulaire, Illkirch, France; [2]Centre National de la Recherche Scientifique, UMR7104, Illkirch, France; [3]Institut National de la Santé et de la Recherche Médicale, U964, Illkirch, France; [4]Université de Strasbourg, Illkirch, France; [5]Division of Structural Studies, MRC Laboratory of Molecular Biology, Cambridge, United Kingdom

**\*For correspondence:** schmidth@igbmc.fr

[†]These authors contributed equally to this work

**Present address:** [‡]Laboratory of Cell Biology, Howard Hughes Medical Institute, The Rockefeller University, New York, United States; [§]Department of Molecular Biology, University of Geneva, Geneva, Switzerland

**Competing interests:** The authors declare that no competing interests exist.

**Abstract** The biogenesis of 60S ribosomal subunits is initiated in the nucleus where rRNAs and proteins form pre-60S particles. These pre-60S particles mature by transiently interacting with various assembly factors. The ~5000 amino-acid AAA+ ATPase Rea1 (or Midasin) generates force to mechanically remove assembly factors from pre-60S particles, which promotes their export to the cytosol. Here we present three Rea1 cryoEM structures. We visualise the Rea1 engine, a hexameric ring of AAA+ domains, and identify an α-helical bundle of AAA2 as a major ATPase activity regulator. The α-helical bundle interferes with nucleotide-induced conformational changes that create a docking site for the substrate binding MIDAS domain on the AAA +ring. Furthermore, we reveal the architecture of the Rea1 linker, which is involved in force generation and extends from the AAA+ ring. The data presented here provide insights into the mechanism of one of the most complex ribosome maturation factors.

DOI: https://doi.org/10.7554/eLife.39163.001

## Introduction

Eukaryotic ribosome assembly is tightly controlled by more than 200 assembly factors to ensure faithful protein synthesis (*Thomson et al., 2013*). During the initial stages of ribosome biogenesis, rRNAs, ribosomal proteins, and assembly factors associate into nucleolar pre-60S particles, which ultimately mature into functional large ribosomal subunits in the cytosol (*Kressler et al., 2010*). The AAA+ (*A*TPases *A*ssociated with various cellular *A*ctivities) family member Rea1 (or Midasin) consists of nearly 5000 amino acids and generates force to mechanically remove assembly factors. Rea1 pulls out the Ytm1 complex (*Tang et al., 2008*; *Sahasranaman et al., 2011*) to promote the transfer of pre-60S particles from the nucleolus to the nucleoplasm (*Bassler et al., 2010*) (*Figure 1A*). Rea1 also removes the assembly factor Rsa4 (*Ulbrich et al., 2009*), which triggers a signalling pathway that ultimately recruits RanGTP to pre-60S particles to export them to the cytosol (*Ulbrich et al., 2009*; *Matsuo et al., 2014*) (*Figure 1A*). Rea1-mediated Rsa4 removal might also indirectly remodel the important H89 rRNA helix of the peptidyltransferase centre into its correct position (*Leidig et al., 2014*; *Bradatsch et al., 2012*; *Baßler et al., 2014*). Despite its crucial importance for pre-60S particle maturation, the Rea1 structure and mechanism have remained largely enigmatic.

Rea1 consists of an N-terminal α-helical domain (NTD), a ring of six AAA+ domains, and the C-terminal tail (*Ulbrich et al., 2009*; *Barrio-Garcia et al., 2016*; *Wu et al., 2016*). The Rea1 tail is subdivided into an α-helical linker region, a D/E-rich region, and a MIDAS (*M*etal-*I*on-*D*ependent

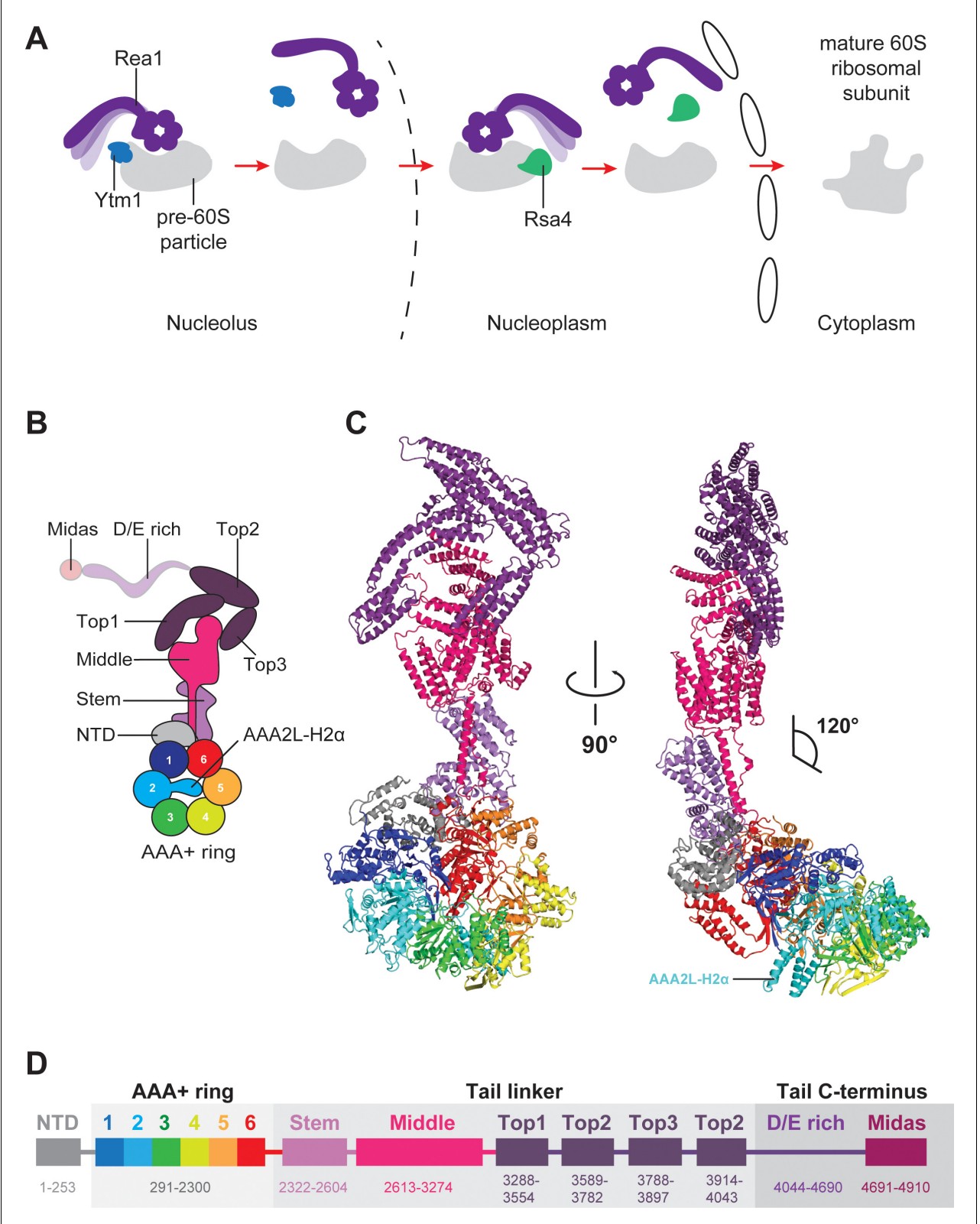

**Figure 1.** Function and structure of Rea1. (**A**) Schematic representation of the reactions catalysed by Rea1. (**B**) Cartoon representation of the Rea1 structure showing the N-terminal domain, the AAA+ ring as well as the stem, middle and top domains of the linker. The top3 domain is an insertion of the top2 domain. The C-terminal D/E-rich region and the MIDAS domain of the tail are not visible as indicated by their transparent representation. (**C**)

*Figure 1 continued on next page*

*Figure 1 continued*

Front (left panel) and side view (right panel) of the Rea1 structure. The AAA2L-H2α insert occupies the central pore of the AAA+ ring. The linker region of the tail emerges at an angle of about 120° from the AAA +ring. (**D**) Colour-coded primary sequence representation of Rea1 domains.

DOI: https://doi.org/10.7554/eLife.39163.002

The following figure supplements are available for figure 1:

**Figure supplement 1.** Quality of NTD-AAA + ring and linker cryoEM maps.

DOI: https://doi.org/10.7554/eLife.39163.003

**Figure supplement 2.** Quality of composite NTD-AAA+ ring linker cryoEM map.

DOI: https://doi.org/10.7554/eLife.39163.004

**Figure supplement 3.** Rea1 mass spectrometry analysis after tryptic digest.

DOI: https://doi.org/10.7554/eLife.39163.005

**Figure supplement 4.** Core architecture of the Rea1 AAA +subunits.

DOI: https://doi.org/10.7554/eLife.39163.006

**Figure supplement 5.** AAA+ ring architecture and nucleotide binding.

DOI: https://doi.org/10.7554/eLife.39163.007

**Figure supplement 6.** Nucleotide binding at the Rea1 AAA1 and AAA6 sites.

DOI: https://doi.org/10.7554/eLife.39163.008

**Figure supplement 7.** Comparison of the Rea1 AAA1 and AAA6 nucleotide-binding sites with the dynein AAA1 and AAA3 nucleotide-binding sites.

DOI: https://doi.org/10.7554/eLife.39163.009

Adhesion *S*ite) domain which is responsible for the interaction with the Ytm1 and Rsa4 substrates (*Bassler et al., 2010*; *Ulbrich et al., 2009*; *Kressler et al., 2012*). Rea1-mediated Ytm1 and Rsa4 removal is ATP dependent (*Bassler et al., 2010*; *Ulbrich et al., 2009*). In analogy to other AAA + family members, it has been proposed that ATP binding and hydrolysis in the AAA +ring is coupled with conformational changes within the Rea1 molecule that generate the force for assembly factor removal (*Ulbrich et al., 2009*; *Kressler et al., 2012*). Pioneering negative stain electron microscopy studies have revealed that the Rea1 tail can extend from the AAA+ ring but is also able to adopt AAA+ ring proximal conformations (*Ulbrich et al., 2009*). The latter tail conformations could bring the MIDAS domain close to its assembly factor substrates when Rea1 is bound to pre-60S particles (*Ulbrich et al., 2009*). It has been proposed that switching between these different tail conformations might produce the force for the removal of Ytm1 and Rsa4 assembly factors (*Kressler et al., 2012*). However, in the absence of high-resolution information, it is unclear what the structure of the Rea1 AAA+ engine looks like, what the molecular architecture of the tail is and how Rea1 binds to its substrates.

## Results

### Overall structure

To provide insights into these open questions, we have determined the *S. cerevisiae* Rea1 structure in complex with ADP by CryoEM, which revealed the NTD, the AAA+ ring, as well as the linker part of the Rea1 tail (*Figure 1B,C*,D). Focused refinement of the NTD-AAA+ ring area and the linker resulted in two cryoEM maps with resolutions of 4.4 Å and 3.9 Å, respectively (*Figure 1—figure supplement 1A–F*, *Figure 1—figure supplement 2* and *Supplementary file 1*). In the NTD-AAA+ ring map, the main-chain is well-resolved in the NTD, AAA1, AAA2, as well as AAA6, and side-chains are occasionally visible (*Figure 1—figure supplement 2*). The AAA3-AAA5 part of the AAA +ring is more flexible, but still allowed docking of AAA+ domains into the map (*Figure 1—figure supplement 2*). The Rea1 linker map revealed side-chains throughout (*Figure 1—figure supplement 2*). The D/E-rich region and the MIDAS domain that follows are not resolved. Mass spectrometry analysis indicates that these regions are present in purified Rea1 (*Figure 1—figure supplement 3*) suggesting their absence is due to the intrinsic flexibility of D/E-rich regions (*Romero et al., 2001*).

### Structure of the Rea1 AAA +ring

The AAA+ ring domains, AAA1-AAA5, are each subdivided into an α/β large domain (AAAL), consisting of five α-helices and five β-strands (H0-H4, S1-S5), and a small domain (AAAS) made up of an

α-helical bundle containing five α-helices (H5-H9) (*Figure 1—figure supplement 4A,B*). All AAALs have β-sheet inserts in H2 as well as in between H4 and S3. In the case of AAA2L, AAA4L, and AAA6L, the H2 β-sheet is extended by α-helical bundles (H2α) of which AAA2L-H2α and AAA4L-H2α are partially disordered (*Figure 1—figure supplement 4A*). Like in other AAA +machines, the Rea1 ring AAA+ domains are arranged as inter-domain modules. Each AAAL, except AAA1L, is tightly associated with the AAAS of the previous domain (*Figure 1—figure supplement 5A*). AAA1L would be expected to interact with AAA6S, but this domain is absent in Rea1 and its place is taken by the NTD.

The nucleotide-binding sites of AAA+ proteins are located at the interfaces between neighbouring AAA+ modules, which carry the conserved Walker-A, Walker-B, and arginine finger motifs responsible for ATP binding and hydrolysis (*Wendler et al., 2012*) (*Figure 1—figure supplement 5B,C*). ATP binding brings these neighbouring modules into close contact to induce ATP hydrolysis. In Rea1, the AAA2, AAA3, AAA4, and AAA5 sites are functional, while AAA1 and AAA6 lack the Walker-B motif involved in ATP hydrolysis (*Kawashima et al., 2016*) (*Figure 1—figure supplement 5D*).

In the structure presented here, the nucleotide binding sites of AAA1 and AAA6 are closed and show densities consistent with a bound nucleotide (*Figure 1—figure supplement 5B* and *Figure 1—figure supplement 6A,B*). We interpreted this bound nucleotide as ADP as there is not enough density to fit a γ-phosphate.

In order to determine the degree of closure at the Rea1 AAA1 and AAA6 sites, we compared them with closed nucleotide binding sites in a crystal structure of the Rea1 related AAA+ family member dynein (*Schmidt et al., 2015*). In the dynein structure, the AAA1 site is trapped in the ATP-hydrolysis transition state due to the presence of ADP.vanadate. All catalytic residues of this site contact the nucleotide and have the right conformation to support hydrolysis, indicating that the site is completely closed (*Schmidt et al., 2015*). We also carried out a comparison with the dynein AAA3 site, which is bound to ADP and, although overall still closed, appears slightly more open than the AAA1 site. The comparison was done by analysing the distance between the H4 α-helix, which carries the arginine finger, and the closest phosphate group of the nucleotide bound to the Walker-A motif of the H1 α-helix (*Figure 1—figure supplement 7A–D*). This distance is around 8 Å in the case of the dynein AAA1 site and around 12 Å in the case of the dynein AAA3 site. The equivalent distances in the Rea1 AAA1 and AAA6 sites are around 10 Å and 8 Å, respectively. For the Rea1 AAA1 site, the degree of closure is intermediate between the completely closed dynein AAA1 site and the 'less closed' dynein AAA3 site. The Rea1 AAA6 appears to be already fully closed in the ADP state.

In contrast to the AAA1 and AAA6 sites, the AAA2 and AAA5 nucleotide-binding sites of the Rea1 AAA+ ring are wide open as indicated by large gaps between modules AAA2/AAA3 and AAA5/AAA6 (*Figure 2A,B,C*). The open state of the latter sites is stabilised by the AAA2L-H2α insert which sits in the AAA+ ring centre. The β-sheet base of AAA2L-H2α is interacting with the AAA1L H2-β-sheet (*Figure 2—figure supplement 1A*) and the α-helices of AAA2L-H2α contact the tips of the AAA3L and AAA5L H2-β-sheets to prevent the closure of the AAA2 and AAA5 sites (*Figure 2B,C* and *Figure 2—figure supplement 1A,B*).

AAA2L-H2α also contacts large parts of the AAA4L H2-β-sheet and enforces a rotation of this domain (*Figure 2—figure supplement 1C* and *Figure 2—figure supplement 2A*). As a consequence, AAA4L H1, which bears the ATP-binding Walker-A motif, moves away from AAA5L H4, which provides the arginine finger motif for ATP hydrolysis at the AAA4 site (*Figure 2D* and *Figure 1—figure supplement 5C*). Furthermore, the AAA2L-H2α interactions with the AAA4L and AAA3L H2-β-sheets also keep the AAA3 site in an open state (*Figure 2A,E* and *Figure 2—figure supplement 1C*).

Our structural analysis suggests that the AAA2L-H2α insert impairs the hydrolytic activity of all functional Rea1 nucleotide binding sites. To provide additional evidence for its inhibitory role, we deleted AAA2L-H2α and determined the ATPase activity of the mutant. In support of our interpretation, the deletion of AAA2L-H2α increased the ATPase rate 10–15 fold compared to wildtype Rea1 (*Figure 2F*). The increase in ATPase activity was specific for the AAA2L-H2α deletion mutant as deleting the AAA6L-H2α insert did not lead to a drastic change in ATPase activity (*Figure 2F*).

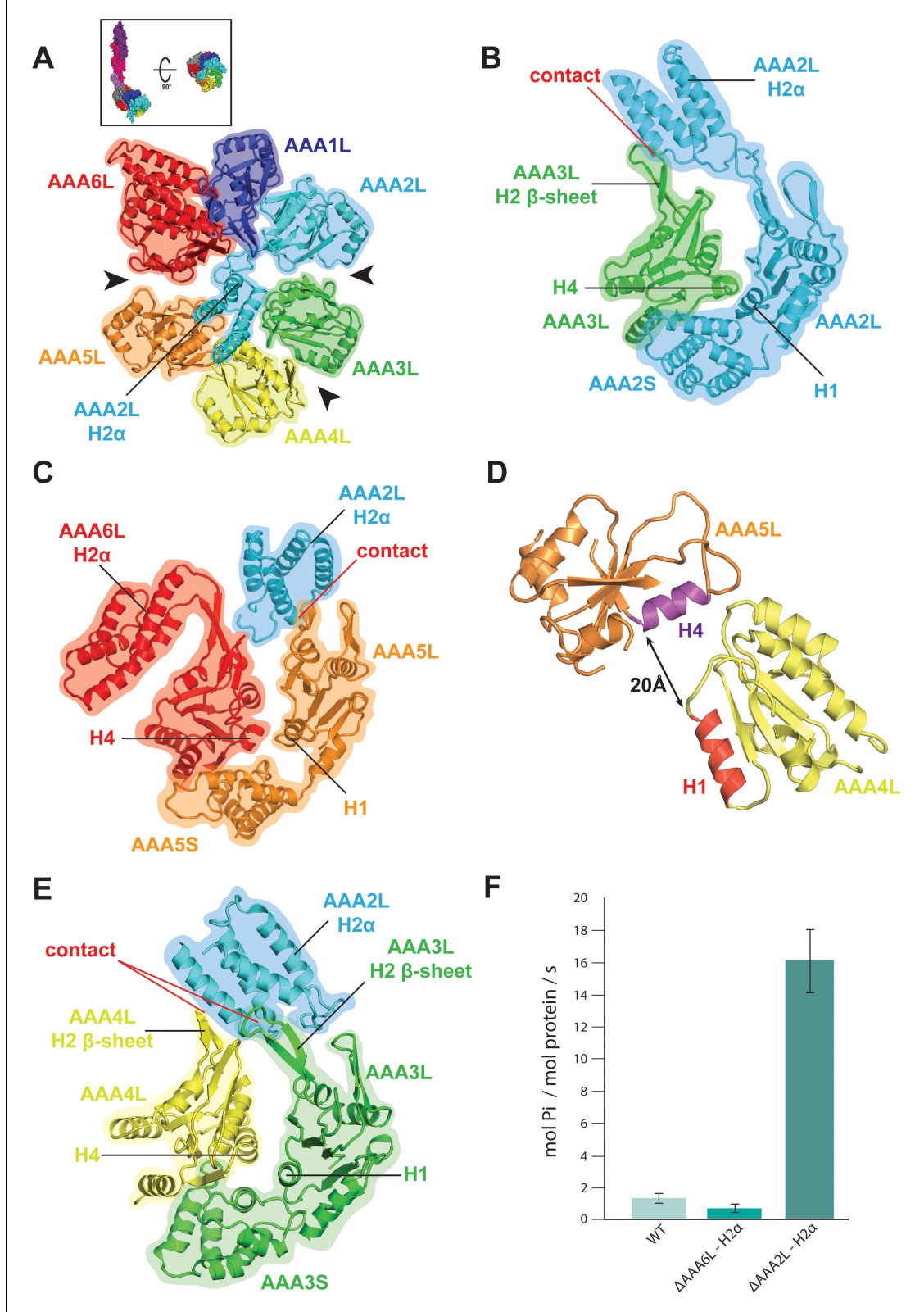

**Figure 2.** Rea1 AAA+ ring architecture and regulation of its ATPase activity. (**A**) Architecture of the Rea1 AAA+ ring. For clarity, only AAAL domains are shown. Gaps exits between AAA2L/AAA3L, AAA3L/AAA4L and AAA5L/AAA6L (black arrows) indicating that nucleotide binding sites AAA2, AAA3 and AAA5 are open. The cartoon in the upper box establishes the view with respect to *Figure 1C* left panel. (**B**) Open AAA2 and (**C**) open AAA5 nucleotide-binding sites. In both cases, contacts between AAA2L-H2α and the AAA3L or AAA5L H2 β-sheets prevent site closure. (**D**) AAA4L is rotated

*Figure 2 continued on next page*

*Figure 2 continued*

with respect to AAA5L (see main text). The rotation moves its ATP-binding H1 α-helix (red) away from the AAA5L H4 α-helix (purple), which carries the R-finger to support ATP-hydrolysis at the AAA4 nucleotide site. The distance is indicated by the black arrow. For clarity parts of AAA5L and AAA4L are not shown. (E) Open AAA3 nucleotide-binding site. The gap between AAA3L and AAA4L is stabilized by contacts between AAA2L-H2α and the H2 β-sheets of AAA3L and AAA4L. (F) ATPase rates for wild type Rea1 (WT) and the AAA6L-H2α (ΔAAA6L-H2α) as well as AAA2L-H2α (ΔAAA2L-H2α) deletion mutants. Deleting AAA6L-H2α decreases the Rea1 ATPase rate, while deleting AAA2L-H2α leads to a 10–15-fold increase in ATPase activity.

DOI: https://doi.org/10.7554/eLife.39163.010

The following figure supplements are available for figure 2:

**Figure supplement 1.** Interactions of AAA2L-H2α insert with H2 β-sheets of AAA1L, AAA3L, AAA4L, and AAA5L.

DOI: https://doi.org/10.7554/eLife.39163.011

**Figure supplement 2.** Rea1 AAA+ ring conformation and its comparison with dynein.

DOI: https://doi.org/10.7554/eLife.39163.012

## Comparison of the Rea1 and dynein AAA+ rings

Rea1 and dynein form a special subclass within the AAA+ family. Most AAA+ members assemble individual AAA+ domains into hexameric rings. In contrast to this more common way of AAA+ ring formation, Rea1 and dynein have their six AAA+ domains concatenated into a single gene (*Garbarino and Gibbons, 2002*). To analyse if there are structural similarities between Rea1 and dynein AAA+ ring geometries, we compared the Rea1 cryoEM structure to a high-resolution crystal structure of the dynein ADP state (*Kon et al., 2012*). The analysis revealed interesting parallels (*Figure 2—figure supplement 2B*).

The overall appearance of the Rea1 AAA+ ring can be described as consisting of two halves adopting an 'open' conformation (*Schmidt, 2015a*). The first half is formed by modules AAA6, AAA1, and AAA2, which tightly associate because of the more closed AAA6 and AAA1 nucleotide binding sites. The second half of AAA+ modules - AAA3, AAA4 and AAA5 - is separated from the first half by the open AAA2 and AAA5 nucleotide binding sites that cause large gaps between modules AAA2/AAA3 and AAA5/AAA6. The internal arrangement of this second half is characterised by a gap between modules AAA3 and AAA4, caused by the open AAA3 site, and the rotation of module AAA4 towards module AAA5 (*Figure 2—figure supplement 2B*).

A similar open AAA+ ring arrangement can be found for the dynein crystal structure (*Schmidt, 2015a*). Here, the AAA2, AAA3, and AAA4 modules form the tightly associated first half due to the closed AAA2 and AAA3 nucleotide binding sites. This first half is separated from the other AAA+ modules by gaps between AAA2/AAA1 and AAA4/AAA5 (*Kon et al., 2012*; *Schmidt, 2015b*). The second half consists of modules AAA5, AAA6, and AAA1 and its internal arrangement is highly similar to the situation in Rea1. The most N-terminal AAA+ module of this half, AAA5, is separated from AAA6 by a gap, and AAA6 is rotated towards AAA1 (*Figure 2—figure supplement 2B*).

The dynein AAA+ ring undergoes nucleotide-dependent conformational changes that cause an open-to-closed transition (*Schmidt, 2015b*). The closed conformation is linked to the remodelling of two AAA+ ring extensions, the dynein linker and the coiled-coil stalk, which are involved in motor force generation and microtubule affinity regulation (*Schmidt et al., 2015*; *Schmidt, 2015b*). In order to accommodate to the closed ring conformation, the AAA6 module rotates towards the AAA + ring centre (*Schmidt et al., 2015*; *Schmidt, 2015b*). Given the similarities in the open AAA+ ring geometries of Rea1 and dynein, especially with respect to the rotated AAA4 and AAA6 modules, it is tempting to speculate that a nucleotide-dependent open-to-closed AAA+ ring transition might also exists in the case of Rea1.

## The AAA+ ring – linker interface and the structure of the Rea1 linker

The AAA+ ring linker interface is formed by the NTD, AAA6L, as well as the stem domain of the linker (*Figure 3A*). Within the AAA+ ring linker interface the NTD acts as a scaffold that helps to stabilise the interactions between AAA6L and the linker stem domain by contacting AAA6L-H2α and the N-terminal stem domain region, which sits like a saddle on top of the NTD. In addition to the linker stem, AAA6L also contacts the linker middle domain (*Figure 3A*), which is located right above the linker stem domain. One of the middle domain α-helices runs along the linker stem towards the

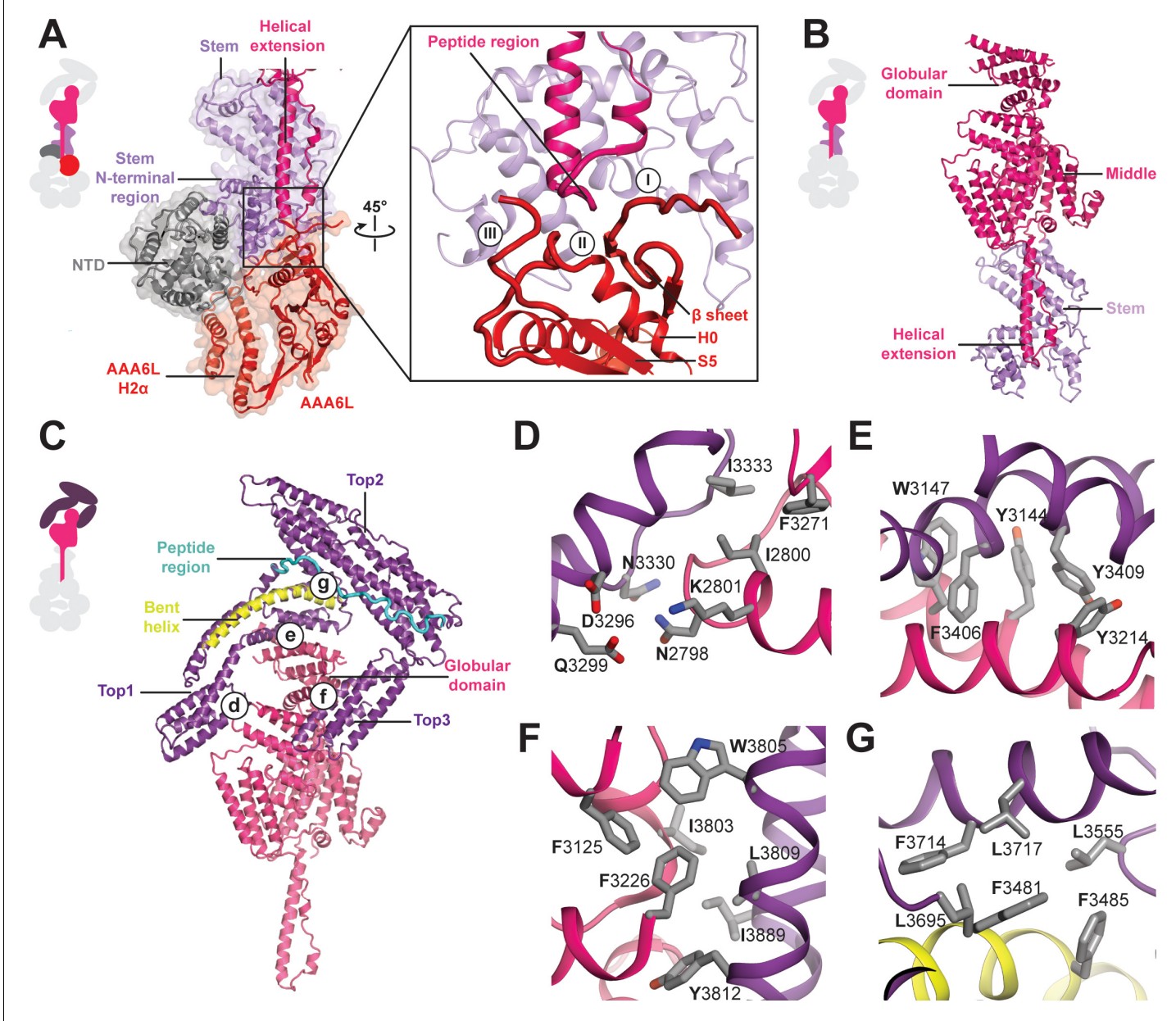

**Figure 3.** Architecture of the Rea1 linker. (**A**) The AAA+ ring linker interface. Left panel: The N-terminal domain (NTD) acts as a scaffold between AAA6L and the linker stem via interactions with AAA6L-H2α and the N-terminal linker stem region. Right panel: Enlarged view of the AAA6L linker interactions. A β-sheet of AAA6L packs against the linker stem. AAA6L is also contacted by an α-helical extension of the linker middle domain. A short peptide region at its tip interacts with: (I) the loop leading to the AAA6L β-sheet, (II) the loop connecting this β-sheet to AAA6L H0 and (III) the loop connecting AAA6L S5 to the linker stem. (**B**) Linker middle domain featuring an α-helical as well as a globular extension. (**C**) The middle domain stabilises the architecture of the linker top. (**D**) Middle domain core and linker top1 interact through charged and hydrophobic residues. The interfaces between the middle domain globular extension and (**E**) top1, as well as (**F**) top3, are mainly hydrophobic. (**G**) Hydrophobic residues dominate the interface between top1 and top2. Top1 features a bent α-helix (yellow) and its connection to top2 is established via a long peptide region (cyan).
DOI: https://doi.org/10.7554/eLife.39163.013

The following figure supplement is available for figure 3:

**Figure supplement 1.** Remodelling of the Rea1 linker with respect to the AAA+ ring.
DOI: https://doi.org/10.7554/eLife.39163.014

AAA+ ring, and a peptide region at its tip contacts the loop leading to a β-sheet that packs against AAA6L (*Figure 3A,B*). Additional contacts exist with the loop connecting this β-sheet to AAA6L H0 and the peptide linker that connects AAA6L S5 to the linker stem domain (*Figure 3A*). AAA6L is the only AAA+ ring part directly contacting the Rea1 linker, which puts it in a key position for communicating ATP induced conformational changes from the AAA+ ring into the linker.

In addition to the stem and middle domain, the Rea1 linker also comprises three α-helical bundles at its top (top1-top3) (*Figure 1B,C,D*). The linker emerges at an angle of roughly 120° from the AAA + ring (*Figure 1C*). The 210 Å length of the complete linker fits well with the ~200 Å AAA+ ring extension that bends towards the pre-60S particle when Rea1 is bound to it (*Ulbrich et al., 2009*), suggesting that the linker is a key element for Rea1 tail remodelling. Previous negative stain EM studies have identified a hinge region within the Rea1 tail that is involved in its remodelling (*Ulbrich et al., 2009*). Comparing 2D projections of our Rea1 structure to this data suggests the pivot point area is located in the region between middle and stem domains of the linker (*Figure 3—figure supplement 1A–C*).

The horseshoe-like arrangement of the three linker top domains is stabilised by the middle domain (*Figure 3C*). Its core contacts top1 via hydrophilic and hydrophobic interactions (*Figure 3D*) and a globular extension of the middle domain interacts with top1 and top3 via predominantly hydrophobic contacts (*Figure 3E,F*). Hydrophobic residues also dominate the interface between the top1 and top2 domains (*Figure 3G*). The various interactions of top1 hold this domain in a strained conformation as indicated by a long, bent α-helix (*Figure 3C*). The connection between the top1 and the top2 domains is established via an extended ~40 aa peptide region between them, which could provide flexibility during linker remodelling (*Figure 3C*). The top3 domain is a four α-helix bundle insertion into the linker top2 domain (*Figure 1D*), and it connects this domain back to the globular extension of the linker middle domain. The central role of the middle domain for the linker top architecture suggest that any conformational change in this linker part will have a major impact on the arrangement of the top domains.

## Insights into the Rea1 nucleotide states

In order to gain insight into nucleotide-dependent Rea1 linker remodelling, we determined the Rea1 cryoEM structure in the presence of the non-hydrolysable ATP analogue AMPPNP to a resolution of 4.3 Å (*Figure 4A*, *Figure 4—figure supplement 1A–D* and *Supplementary file 1*). The Rea1 AMPPNP state is highly similar to the ADP state. The Rea1 ADP structure can be docked into the cryoEM density with minimal adjustments (*Figure 4—figure supplement 1E*). The fact that there were no significant conformational changes in the AAA+ ring compared to the ADP state reinforces our interpretation of the AAA2L-H2α insert as an auto-inhibitory element that prevents movement between the AAA+ modules of the ring. Since there was no evidence for linker remodelling in the AMPPNP state, we decided to investigate the alternative APO and ATP nucleotide states by negative stain electron microscopy. However, in these states we also did not observe linker remodelling (*Figure 4—figure supplement 2A,B*).

We hypothesised that linker remodelling might be obscured by the auto-inhibitory AAA2L-H2α insert and carried out negative stain electron microscopy investigations on a Rea1_ΔAAA2L-H2α mutant in the APO, AMPPNP, ATP, and ADP states. Like in the case of Rea1, there was no evidence for a large scale linker remodelling event and the angle between AAA+ ring and linker remained around 120° (*Figure 4—figure supplement 3A–D*).

Although we did not observe linker remodelling, there were noticeable differences for the AAA + ring in case of the Rea1_ΔAAA2L-H2α mutant. In the APO and AMPPNP states, the AAA+ ring was extended by an additional EM density (*Figure 4—figure supplement 3A,B*). Docking the Rea1_ΔAAA2L-H2α APO structure into a recently published cryoEM structure of an Rsa4 containing Rea1-pre-60S particle complex (*Barrio-Garcia et al., 2016*) would place the extra AAA+ ring density directly above the N-terminal MIDO domain of Rsa4, which is known to bind to the Rea1 MIDAS domain (*Ulbrich et al., 2009*) (*Figure 4—figure supplement 4A,B*). The docking revealed that that the potential MIDAS domain density would overlap with a so far unidentified density in the Rea1-pre-60S particle map (*Figure 4—figure supplement 4A,B*). We hypothesised that the extra AAA + ring density is the Rea1 MIDAS domain. To confirm our interpretation, we deleted the MIDAS domain in the ΔAAA2L-H2α background and investigated the APO and AMPPNP states of the Rea1_ΔAAA2L-H2α_ΔMIDAS double deletion mutant by negative stain electron microscopy. In

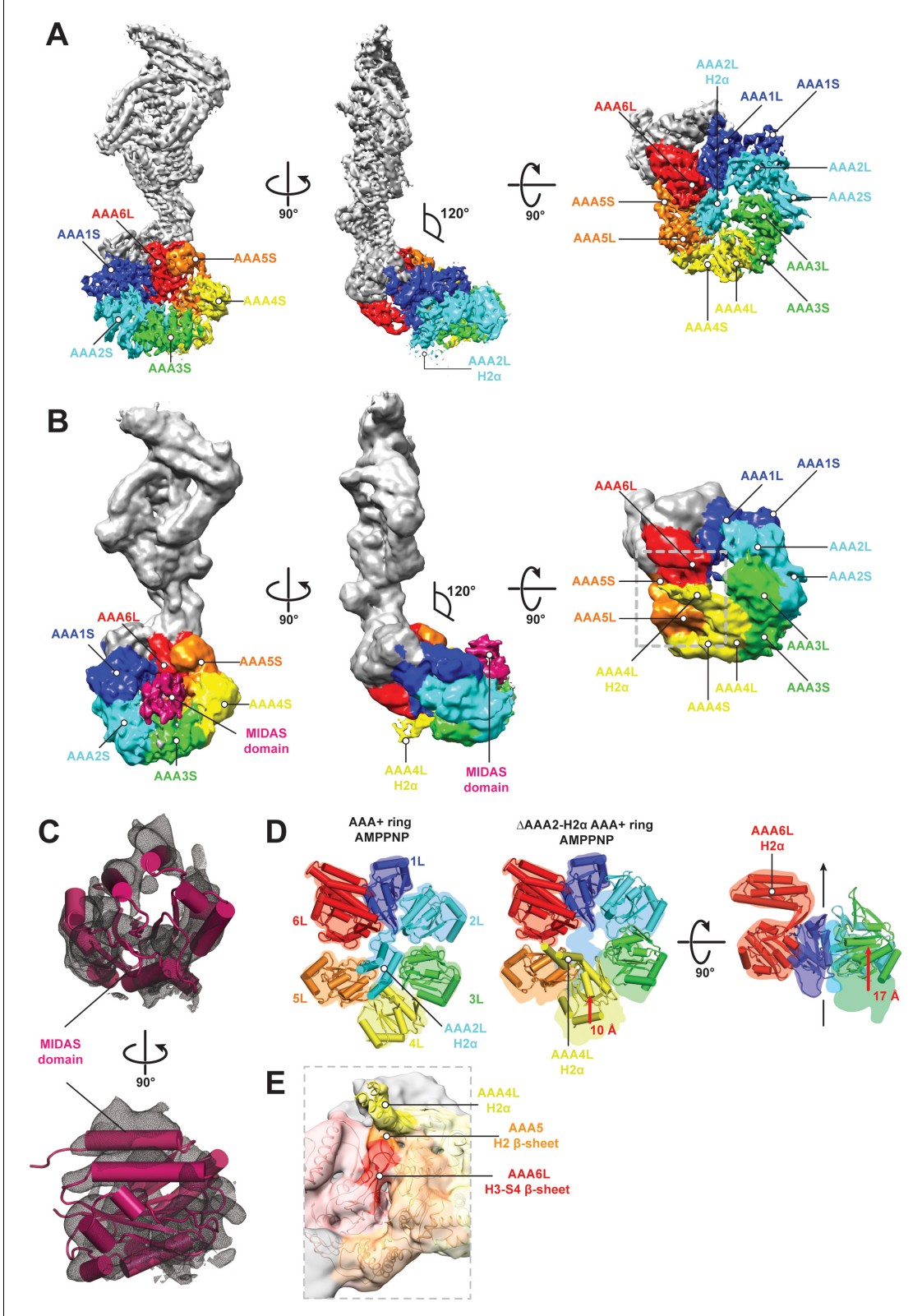

**Figure 4.** CryoEM structures of the Rea1 and Rea1_ΔAAA2L-H2α AMPPNP states. (**A**) Front (left panel) and side (middle panel) view of the Rea1 AMPPNP state. The relative orientation of the linker with respect to the AAA+ ring has not changed. The AAA2L-H2α insert sits in the middle of the AAA+ ring (right panel), (**B**) Front (left panel) and side (middle panel) view of the Rea1_ΔAAA2L-H2α AMPPNP state. The angle between the linker and the AAA+ ring remains at 120°. A cryoEM density consistent with a MIDAS domain is visible above the AAA2S and AAA3S domains. The density for

*Figure 4 continued on next page*

*Figure 4 continued*

AAA2L-H2α has disappeared and the AAA4L-H2α insert is visible in the middle of the AAA+ ring (right panel) (C) Enlarged view of the MIDAS domain density demonstrating the fit with the integrin alpha-L MIDAS domain (purple cartoon, PDB-ID: 1T0P). Upper panel: front view, lower panel: side view. (D) Side-by-side comparison of the AAA+ rings of the Rea1 (left panel) and the Rea1_ΔAAA2L-H2α (middle and right panel) AMPPNP states. For clarity, only the AAAL domains (cartoon representation) are shown. The colour-coded outline of the AAAL positions of the Rea1 AMPPNP state (left panel) is superimposed on the AAAL positions of the Rea1_ΔAAA2L-H2α AMPPNP state (middle and right panels). For the right panel, AAA4L, and AAA5L were removed to better demonstrate the shift of AAA3L. The black arrow runs through the ring centre and indicates the direction towards the H2α insert side of the ring. Red arrows indicate the shift of the AAAL core. (E) Side view of the area marked by the grey dotted box of the right panel in (B) in transparent surface and cartoon representation. The AAA4L-H2α insert is in close proximity to the AAA5L H2 β-sheet and the AAA6L H3-S4 H2-β-sheet.

DOI: https://doi.org/10.7554/eLife.39163.015

The following figure supplements are available for figure 4:

**Figure supplement 1.** Quality of the Rea1 AMPPNP map and comparison of the Rea1 ADP and AMPPNP structures.

DOI: https://doi.org/10.7554/eLife.39163.016

**Figure supplement 2.** Negative stain electron microscopy reconstructions of Rea1 APO and ATP states.

DOI: https://doi.org/10.7554/eLife.39163.017

**Figure supplement 3.** Negative stain electron microscopy reconstructions of Rea1_ΔAAA2L-H2α APO, AMPPNP, ATP and ADP states.

DOI: https://doi.org/10.7554/eLife.39163.018

**Figure supplement 4.** The additional density at the AAA+ ring of the Rea1_ΔAAA2L-H2α APO and AMPPNP states is the MIDAS domain.

DOI: https://doi.org/10.7554/eLife.39163.019

**Figure supplement 5.** CryoEM structure of the Rea1_ΔAAA2L-H2α AMPPNP state.

DOI: https://doi.org/10.7554/eLife.39163.020

**Figure supplement 6.** Comparison of Rea1 and Rea1_ΔAAA2L-H2α AMPPNP states.

DOI: https://doi.org/10.7554/eLife.39163.021

support of our interpretation as the MIDAS domain, the extra density had disappeared (*Figure 4— figure supplement 4C–F*). In contrast to the APO and AMPPNP states, there was no evidence for MIDAS domain AAA+ ring docking in the ATP or ADP states (*Figure 4—figure supplement 3C,D*).

In order to gain insight into the molecular basis for MIDAS domain docking, we decided to determine the cryoEM structure of the Rea1_ΔAAA2L-H2α AMPPNP state. The overall resolution of the reconstruction was 7.8 Å. The reconstruction revealed a density consistent with a MIDAS domain in-between the AAA3 and AAA4 modules, above AAA2S and AAA3S, respectively (*Figure 4B,C*, *Figure 4—figure supplement 5A–D* and *Supplementary file 1*). As in the case of the Rea1_ΔAAA2L-H2α APO state, docking of this structure into the Rsa4 containing pre-60S structure would place the MIDAS domain in close contact with its Rsa4 substrate (*Figure 4—figure supplement 5E*). Since the auto-inhibitory AAA2L-H2α insert has been deleted, the presence of AMPPNP is able to induce movements between the AAA+ modules of the AAA+ ring. Comparing the AMPPNP cryoEM structures of the Rea1_ΔAAA2L-H2α mutant and Rea1 allowed us to analyse these conformational changes and investigate how they contribute to the formation of the MIDAS domain binding site. The NTD, AAA1, AAA6 and the linker can be superimposed as a single rigid body, whereas the AAA2-AAA5 part has moved (*Figure 4—figure supplement 6A*). The most drastic difference is apparent for the AAA4 module. Since the AAA2L-H2α insert is no longer pushing against AAA4L, the AAA4 module is free to move towards the central pore of the ring (*Figure 4D*). Two rod-shaped cryoEM densities emerge from AAA4L that were not present in the Rea1 AMPPNP map. We interpret these densities as two α-helices of the previously disordered AAA4L-H2α insert (*Figure 4B,E*). The AAA4L-H2α insert occupies the area taken by AAA2L-H2α in the Rea1 AMPPNP map (*Figure 4D*). It is located in close proximity to the AAA5L H2 and AAA6L H4-S3 β-sheets (*Figure 4E*), which contact each other because the AAA5 module has moved towards the AAA6 module (*Figure 4D*). Contacts between AAA4L-H2α and the β-sheets inserts of AAA5L and AAA6L might help stabilising the movement of the AAA4 module towards the AAA+ ring centre.

Another prominent conformational change is evident for the AAA3 module. It has moved around 17 Å vertically towards the H2α insert side of the AAA+ ring (*Figure 4D*). The consequence of the combined movements of the AAA4 and AAA3 modules is that their small domains, AAA3S and AAA2S, adopt the right orientation with respect to each other to from the MIDAS domain binding site (*Figure 4—figure supplement 6B*).

# Discussion

The first Rea1 high-resolution structure has allowed us to gain important insights into the architecture and regulation of this essential molecular machine. The most prominent feature of the Rea1 AAA+ ring is the AAA2L-H2α insert that sits like a plug in the central pore of the AAA+ ring. In other AAA+ family members, such as the mitochondrial protease YME1, the disaggregase Hsp104, or the unfoldase VAT, the AAA+ ring pore interacts with their respective substrates (*Gates et al., 2017*; *Puchades et al., 2017*; *Ripstein et al., 2017*). Instead of binding to the Ytm1 or Rsa4 substrates, the Rea1 AAA+ ring centre hosts a structural element that regulates the ATPase activity. Its central position allows the AAA2L-H2α insert to influence all conserved Rea1 ATP hydrolysis sites in parallel. The structural and ATPase data presented here suggest that Rea1 exists in an auto-inhibited state with impaired hydrolytic activity at sites AAA2-AAA5. However, previous work has established the importance of ATP hydrolysis at these sites for yeast growth (*Kawashima et al., 2016*). Furthermore, the AAA3 site has also been directly implicated in Rsa4 assembly factor removal and pre-60S particle export to the cytosol (*Barrio-Garcia et al., 2016*). This raises the question how Rea1 might be activated to carry out its essential function. Recent studies have established that the AAA2L-H2α insert interacts with the Rix1 component of pre-60S particles to recruit Rea1 to its substrates (*Barrio-Garcia et al., 2016*). We suggest here that this binding event relocates the AAA2L-H2α insert from the AAA+ ring pore to stimulate the Rea1 ATPase activity (*Figure 5*). In support of this idea, a recent cryoEM structure of a Rea1-pre-60S particle complex has revealed a density extending from the Rea1 AAA+ ring centre towards Rix1 that was interpreted as the AAA2L-H2α insert (*Barrio-Garcia et al., 2016*).

A key question of the Rea1 mechanism is how ATP induced conformational changes in the AAA + ring drive the remodelling of the linker to generate force for assembly factor removal. Our analysis of the AAA+ ring – linker interface identified interactions between AAA6L, the linker stem and the helix extension of the middle domain (*Figure 5*). It is conceivable that a movement of AAA6L during ATP hydrolysis in the ring is transferred to the linker middle domain via the interaction between AAA6L and the middle domain helix extension. A subsequent shift in the position of the middle domain, with the linker stem as pivot point, would be suitable to disrupt the architecture of the linker top to trigger a large-scale remodelling event.

We determined Rea1 and Rea1_ΔAAA2L-H2α electron microscopy structures in the APO, AMPPNP, ATP, and ADP nucleotide states to get insight into Rea1 linker remodelling. However, we were not able to observe the expected large-scale rearrangement of the linker with respect to the AAA+ ring. The relative orientation between these Rea1 parts remained essentially the same in all structures. We can only speculate about the reasons for this absence of linker remodelling. One possibility would be that linker remodelling relies on critical interactions between Rea1 and the pre-60S particle so that it can only be observed when Rea1 is bound to pre-60S particles. Evidence for this hypothesis is provided by the Rea1-pre-60S particle cryoEM structure (*Barrio-Garcia et al., 2016*). The AAA+ ring cryoEM density is much stronger than the linker cryoEM density. This suggests that the linker becomes flexible with respect to the ring when Rea1 is bound to pre-60S particles. Another possibility to explain the lack of alternative linker conformations are yet unidentified protein co-factors that participate in linker remodelling.

Our work on the Rea1 nucleotide states revealed the existence of a MIDAS domain binding site on the AAA+ ring. At the pre-60S particle, this binding site would place the MIDAS domain in direct contact with its Rsa4 substrate. The Rea1 and Rea1_ΔAAA2L-H2α AMPPNP structures provide strong evidence for the idea that the AAA2L-H2α insert interferes with the nucleotide-dependent conformational changes of the AAA3 and AAA4 modules that create the MIDAS domain binding site.

The D/E-rich region between the linker top2 domain and the MIDAS domain was not visible in our Rea1_ΔAAA2L-H2α AMPPNP map suggesting it remains flexible even when the MIDAS domain is fixed to the AAA+ ring. We observed the binding of MIDAS domain only in the APO and AMPPNP states, but not in the ATP and ADP states. This suggests that ATP hydrolysis correlates with the disappearance of the MIDAS domain binding site. When Rea1 is bound to pre-60S particles, this event might be accompanied by the removal of the Rsa4 assembly factor. The flexible D/E region between the MIDAS domain and the linker raises the question of how force can be transmitted to the MIDAS domain by a linker remodelling event. Since the Rsa4-bound MIDAS domain would be fixed on the

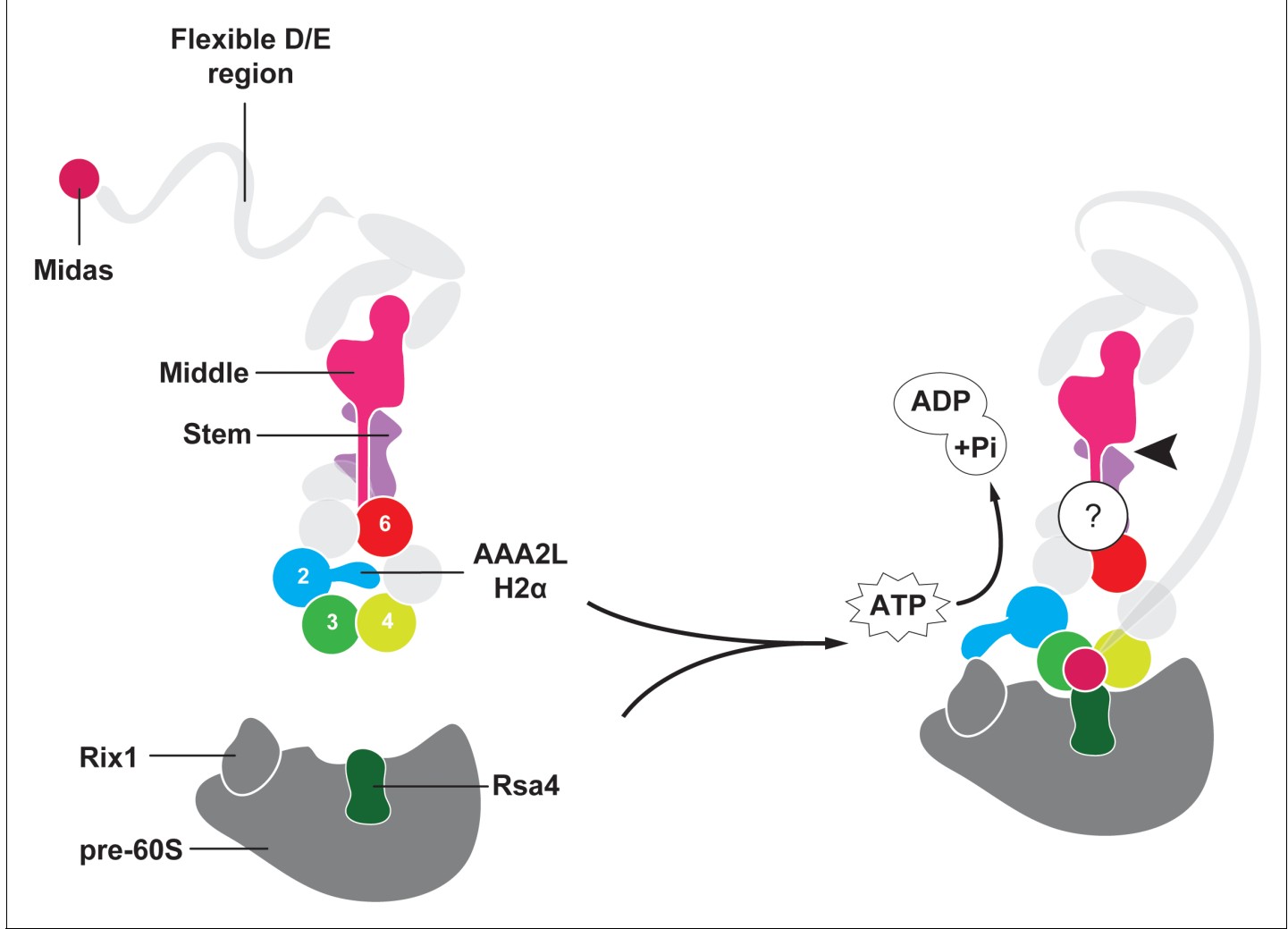

**Figure 5.** Model for Rea1 mediated Rsa4 assembly factor removal. In the absence of pre60S particles the Rea1 ATP-hydrolysis activity is inhibited by AAA2L-H2α. When Rea1 binds to pre60S particles, AAA2L-H2α relocates towards Rix1, and the Rea1 ATPase activity is stimulated. AAA2L-H2α removal allows nucleotide dependent conformational changes in the ring that cause a shift in the positions of the AAA3 and AAA4 modules and result in the recruitment of the MIDAS domain. The AAA+ ring binding site brings the MIDAS domain in close contact with its Rsa4 substrate. AAA6L of the AAA6 module has the ideal position to communicate ATP-dependent conformational changes of ring into the linker via its interface with the linker stem and the helix extension of the linker middle domain. The interface between linker stem and middle domain (black arrow head) might act as a pivot point during linker remodelling. The rearrangement of the linker during remodelling, and how it produces force to remove the Rsa4 engaged MIDAS domain from the AAA+ ring remains unknown (question mark).

DOI: https://doi.org/10.7554/eLife.39163.022

AAA+ ring, a linker remodelling event might simply stretch the flexible D/E region and create the force for Rsa4 removal.

In summary, we presented the first Rea1 high-resolution structure and revealed the intricate sub-domain architecture of the Rea1 linker. The AAA2L-H2α insert inhibits Rea1 ATPase activity as well as the formation of the MIDAS domain binding site on the AAA+ ring. Rea1 is recruited to pre-60S particles through an interaction between the AAA2L-H2α insert and Rix1. This binding event might relocate the AAA2L-H2α insert from the AAA+ ring centre to Rix1. Such a relocation event would stimulate the Rea1 hydrolysis activity and allow nucleotide driven conformational changes in the AAA+ ring that lead to the formation of the MIDAS domain binding site (*Figure 5*).

# Materials and methods

## Key resources table

| Reagent type (species) or resource | Designation | Source or reference |
|---|---|---|
| Strain, strain background (*S. cerevisae*) | JD1370 | DOI: 10.1126/science.1212642 |
| Chemical compound | 1,4-dithiotreitol (DTT) | Thermo Fisher SCIENTIFIC |
| Chemical compound | ATP | ACROS Organics |
| Chemical compound | AMPPNP | Jena biosciences |
| Chemical compound | ADP | SIGMA-ALDRICH |
| Chemical compound | Roche cOmplete, EDTA-free Protease Inhibitor | SIGMA-ALDRICH |
| Chemical compound | Dimethyl sulfoxide (DMSO) | SIGMA-ALDRICH |
| Chemical compound | Phenylmethyl sulfonyl fluoride (PMSF) | SIGMA-ALDRICH |
| Chemical compound | Yeast Nitrogen Base without Amino acids | Formedium |
| Chemical compound | D(+) - Glucose | Formedium |
| Chemical compound | D(+) - Galactose | Formedium |
| Chemical compound | CSM, -Ura | Formedium |
| Chemical compound | Triton X-100 | SIGMA-ALDRICH |
| Commercial assay or kit | EnzChek Phosphate Assay Kit | Thermo Fisher SCIENTIFIC |
| Software, algorithm | Adobe Photoshop version 16.0.3 (for figure preparation) | Adobe Systems, Inc. N/A |
| Software, algorithm | PyMOL(TM) 2.0.6 Schrodinger LLC | https://pymol.org/edu/?q=educational/ |
| Software, algorithm | Chimera *Pettersen et al., 2004* | https://www.cgl.ucsf.edu/chimera/download.html |
| Software, algorithm | Gautomatch | https://www.mrc-lmb.cam.ac.uk/kzhang/Gautomatch/ |
| Software, algorithm | Serial EM *Mastronarde, 2005* | http://bio3d.colorado.edu/SerialEM |
| Software, algorithm | MotionCor2 *Zheng et al., 2017* | http://msg.ucsf.edu/em/software/motioncor2.html |
| Software, algorithm | RELION 2.0 *Kimanius et al., 2016* | http://www2.mrc-lmb.cam.ac.uk/relion |
| Software, algorithm | COOT *Emsley and Cowtan, 2004* | http://www2.mrc-lmb.cam.ac.uk/personal/pemsley/coot |
| Software, algorithm | PHENIX *Adams et al., 2010* | https://www.phenix-online.org |

## Protein expression and purification

*Saccharomyces cerevisiae* Rea1 was amplified from genomic DNA using standard molecular biology methods. The construct was cloned into a pYES2 vector (Thermofisher) modified to introduce an N-terminal tandem Protein-A tag followed by two preScission protease cleavage sites, GFP, and two TEV protease cleavage sites. For the Rea1_ΔAAA2L-H2α and Rea1_ΔAAA6L-H2α deletion mutants, residues 692–813 or 2092–2196 were replaced by a short SG-linker. To create the Rea1_ΔAAA2L-H2α_ΔMIDAS mutant, residues 4702–4910 were deleted in addition to the residue range 692–813. Rea1 plasmids were transformed into yeast JD1370 using the Ura selection marker. For large-scale expression, 60 ml pre-cultures (2% glucose, CSM-Ura minimal medium, (Formedium)) were used to inoculate 6 × 1L expression medium (2% galactose, CSM-Ura minimal medium (Formedium)). Flasks were incubated at 30°C and 240 rpm for 12–24 hr. Cells were pelleted at 4000 rpm, resuspended in water, and flash frozen in liquid nitrogen. The frozen pellets were blended and resuspended in lysis buffer (30 mM HEPES-NaOH pH 7.4, 50 mM K-Acetate, 2 mM Mg-Acetate, 0.2 mM EGTA, 1 mM DTT, 3 mM ATP, 0.2 mM PMSF). All the following procedures were carried out at 4°C. The lysate was centrifuged at 60000 rpm for 40 min. The supernatant was collected and incubated with IgG sepharose beads (GeHealthcare) for 1 hr. Beads were washed with buffer A (50 mM HEPES-NaOH pH 7.4, 50 mM K-Acetate, 2 mM Mg-Acetate, 0.2 mM EGTA, 10% Glycerol, 100 mM KCl, 1 mM DTT, 3 mM ATP) and incubated with buffer B (50 mM TRIS-HCl pH 8.0, 150 mM K-Acetate, 2 mM Mg-Acetate, 1 mM EGTA, 10% Glycerol, 1 mM DTT). The protein was cleaved off the beads with preScission protease overnight. The flow through was combined and concentrated. The concentrated sample was loaded on a superose 6 column (GE Healthcare Life Sciences) equilibrated with buffer B. For EM grid preparation, the glycerol component of buffer B was omitted. The peak fractions were used for the ATPase assays and the EM grid preparations.

## ATPase assays

For all ATPase assays, the EnzChek Phosphate Assay Kit (Molecular Probes) was used according to the recommendations of the supplier. The final reaction volume was 150 µl consisting of 30 µl 5x assay buffer (150 mM HEPES-NaOH pH 7.2, 10 mM Mg-Acetate, 5 mM EGTA, 50% Glycerol, 5 mM DTT), 30 µl MESG (EnzCheck substrate), 1.5 µl PNP (purine nucleoside phosphorylase), 10 µl Mg-ATP, and protein. The final protein concentration was 250 nM. Experiments were carried out on a GENios spectrophotometer (TECAN). The ATPase assays for Rea1 and the Rea1_ΔAAA2L-H2α deletion mutant were done in triplicate. The Rea1_ΔAAA6L-H2α ATPase assays were done in duplicate. The ATPase rates were 1.1 ± 0.13, 0.5 ± 0.1, and 16.1 ± 2.0 mol Phosphate/mol Rea1/s for Rea1, Rea1_ΔAAA6L-Hα, and Rea1_ΔAAA2L-H2α, respectively.

## CryoEM sample preparation and data acquisition

Gel filtration fractions of Rea1 and Rea1_ΔAAA2L-H2α were diluted to 1 mg/ml using buffer B (without glycerol) and ADP or AMPPNP were added to a final concentration of 3 mM. 3 µl of sample were applied to glow-discharged holey carbon grids (Quantifoil Cu R2/2, 300-square-mesh) that were subsequently blotted for 7–8 s and flash frozen in liquid ethane using a manual plunger. All cryoEM grid preparations were carried out at 4°C. All data were collected on an FEI Titan Krios equipped with a Cs corrector and a Gatan K2-Summit detector (300 kV, 35–38 frames, 7–8 s exposure, 1.09 Å/pixel,~45–50 e̅/Å$^2$) using a slit width of 20 eV on a GIF-quantum energy filter (Gatan). All images were recorded in super-resolution counting mode using the automated data collection software Serial EM (*Mastronarde, 2005*) with a defocus range of 1.8 to 3.4 µm. In the case of the Rea1_ΔAAA2L-H2α AMPPNP state, a Volta phase plate was used in combination with a target defocus of 0.5 µm. The irradiated area on the VPP was changed every hour.

## Image processing

Motion correction and dose-weighting were performed using MotionCor2 (*Zheng et al., 2017*). CTF parameters were estimated using Gctf (*Zhang, 2016*). Subsequent processing was done using Relion-2.0 (*Kimanius et al., 2016*) unless stated otherwise. Micrographs were first manually examined to remove images with significant uncorrected drift, large amount of contamination, a large astigmatism, extreme defocus values (<1.2 µm or >5 µm), or abnormal Fourier patterns. Micrograph

quality evaluation in the case of the Rea1_ΔAAA2L-H2α AMPPNP phase plate data set was done according to *von Loeffelholz et al. (2018)*.

For the Rea1 ADP dataset a small set of particles was manually picked and subjected to reference-free 2D classification. A selection of 2D class averages representing different views of Rea1 was selected, centred using the Relion shift_com function, and used as references for autopicking by Gautomatch (K. Zhang, MRC Laboratory of Molecular Biology, Cambridge, UK). Autopicked particles were cleaned by several cycles of 2D classification. Subsequently, cleaned particles were subjected to 3D refinement. The initial Rea1 model was obtained using the *ab-initio* modelling function as implemented in Relion 2.1. The output X and Y origin information of the 3D refinement was used to obtain more accurate coordinates of the particles. The re-centred particles were subjected to another round of 3D refinement. The obtained map was divided into two parts covering the Rea1 linker region and the NTD-AAA+ ring region. We conducted focused 3D classification and 3D refinement of these individual parts as described (*Nguyen et al., 2016*). The final reconstructions of the NTD-AAA+ ring region and the linker were based on 35671 and 432556 particles, respectively. To aid the interpretation of the obtained maps, sharpening was carried out by applying a negative B-factor that was either estimated using automated procedures within Relion or manually set parameters. The Rea1 AMPPNP state was also reconstructed with Relion 2.1. The Rea1 ADP state map was low pass filtered and used as an initial reference for the first 3D refinement step. Particles were aligned using the output of the initial refinement as a reference followed by a focused refinement of the Rea1 linker part. The subsequent 3D classification was performed post alignment using a large elliptic featureless mask encompassing the Rea1 AAA+ ring to evaluate potential movement between the linker and the AAA+ ring. In all analysed classes, the linker-AAA+ ring angle remained around 120°. The final Rea1 AMPPNP map was reconstructed from a class with 55442 particles. The Rea1_ΔAAA2L-H2α AMPPNP data set was processed with Relion 3.0 using standard procedures including *ab-initio* reference building by Stochastic Gradient Descent. The final Rea1_ΔAAA2L-H2α AMPPNP map was reconstructed from 20724 particles.

## Model building and refinement

All model buildings were done in Coot (*Emsley and Cowtan, 2004*). In the case of the NTD-AAA + ring of the Rea1 ADP map, polyalanine models of the dynein AAA1L and AAA1S domains (PDB-ID: 4RH7) were initially docked into the AAA+ ring and subsequently modified according to the quality of the map. Other parts of this map were built by placing standardised α-helices and tracing the main-chain between them. Occasionally bulkier side-chains were visible that were included in the final model. The quality of the Rea1 linker map allowed de-novo building of a side-chain model. Phenix real space refine was used to refine the NTD-AAA+ ring and the linker model (*Adams et al., 2010*). To create the Rea1 ADP AAA+ ring – linker composite model, both structures were aligned on the stem domain which was visible in both maps. For the Rea1 AMPPNP model, the Rea1 ADP model was split into the NTD-AAA+ ring linker stem and linker middle domain – linker top regions. Both regions were docked into the Rea1 AMPPNP map as two separate rigid-bodies. This initial docking already led to a close fit. Minimal rigid body fit adjustments were done for the AAA2, AAA3, AAA4, and AAA5 modules of the AAA+ ring. To create the Rea1_ΔAAA2L-H2α AMPPNP model, the NTD, the AAA1, and AAA6 modules, as well as the complete linker of the Rea1 ADP structure, were docked as a single rigid body into the Rea1_ΔAAA2L-H2α AMPPNP map. The AAA2, AAA3, AAA4, and AAA5 modules of the Rea1 ADP structure were docked as individual rigid bodies into the Rea1_ΔAAA2L-H2α AMPPNP map. All figures were prepared in Pymol (The PyMOL Molecular Graphics System, Version 1.2r3pre, Schrödinger, LLC) or Chimera (*Pettersen et al., 2004*). The programme Eman2 (*Tang et al., 2007*) was used for 2D projections.

## Negative stain sample preparation and data acquisition

Rea1 or the Rea1_ΔAAA2L-H2α deletion mutant in buffer B without glycerol were diluted to a final concentration of 45 nM. AMPPNP, ATP, or ADP were added to reach a final concentration of 3 mM. Negative-stain electron microscopy was performed on plasma-cleaned carbon film on 400-square-mesh copper grids (Electron Microscopy Sciences). 3 μl sample was applied to the grids that were subsequently stained with 2% (w/v) uranyl acetate. Data collection was done on an FEI tecnai G2 operated at 200kV and equipped with a Gatan Ultrascan 2K*2K CCD camera. Data were collected at

~1 µm underfocus, with a pixel size of 3.629 Å and an estimated dose of 25 electrons / Å$^2$ during 1 s exposures. SerialEM was used for semiautomatic data acquisition. Around 8000 particles per data set were manually picked and processed with Relion using standard procedures. In order to not bias the orientation of the linker with respect to the AAA+ ring, the initial model consisted of the low pass filtered NTD-AAA+ ring map rescaled to the correct pixel size.

## Data availability

The atomic coordinates for the Rea1 AAA+ ring and the Rea1 linker in the ADP state have been deposited with PDB IDs 6HYP and 6HYD, respectively. The accession codes for the Rea1 and Rea1_ΔAAA2L-H2α models in the AMPPNP state are 6I26 and 6I27, respectively. The accession codes for the cryoEM maps of the Rea1 AAA +ring and the Rea1 linker in the ADP state are EMD-0309 and EMD-0308, respectively. The accession code for the unsharpened cryoEM map of the Rea1 AAA +ring in the ADP state is EMD-0330. The cryoEM maps of the Rea1 and Rea1_ΔAAA2L-H2α AMPPNP states have the EMD accession codes EMD-0328 and EMD-0329, respectively.

## Acknowledgements

We gratefully acknowledge Julio Ortiz and Tat Cheung Cheng for their help during cryoEM data collection. This research was supported by ATIP-avenir (CDP-0B1INSB-HS9ADO1051) as well as LabEx start up (ANR-10-LABEX-30-HS) grants to HS and a Region Grand Est jeunes chercheurs fellowship to PS. HS also acknowledges the support of the MRC Laboratory of Molecular Biology during the initial stages of the project. This study was further supported by the grant ANR-10-LABX-0030-INRT, a French State fund managed by the Agence Nationale de la Recherche under the frame program Investissements d'Avenir ANR-10-IDEX-0002–02. The authors acknowledge the support and the use of resources of the French Infrastructure for Integrated Structural Biology (FRISBI) ANR-10-INBS-05 and of Instruct-ERIC. We thank Andrew Carter, Patrick Schultz, Albert Weixlbaumer, Alastair McEwen and Sandrine Morlot for their invaluable comments on the manuscript.

## Additional information

### Funding

| Funder | Grant reference number | Author |
| --- | --- | --- |
| Région Grand Est | jeunes chercheurs fellowship | Piotr Sosnowski |
| ATIP-Avenir | CDP 0B1INSB-HS-9ADO1051 | Helgo Schmidt |
| Labex | ANR-10-LABEX-30-HS | Helgo Schmidt |

The funders had no role in study design, data collection and interpretation, or the decision to submit the work for publication.

### Author contributions

Piotr Sosnowski, Conceptualization, Data curation, Formal analysis, Validation, Investigation, Methodology, Writing—original draft, Writing—review and editing; Linas Urnavicius, Andreas Boland, Johan Busselez, Data curation, Software, Formal analysis, Methodology; Robert Fagiewicz, Data curation, Software, Formal analysis, Writing—original draft, Writing—review and editing; Gabor Papai, Data curation, Software, Methodology; Helgo Schmidt, Conceptualization, Resources, Data curation, Software, Formal analysis, Supervision, Funding acquisition, Validation, Methodology, Writing—original draft, Project administration, Writing—review and editing

### Author ORCIDs

Piotr Sosnowski (ID) http://orcid.org/0000-0003-4902-9560
Andreas Boland (ID) http://orcid.org/0000-0003-1218-6714

Johan Busselez  http://orcid.org/0000-0002-4078-1265
Helgo Schmidt  http://orcid.org/0000-0002-8004-8316

**Decision letter and Author response**
Decision letter https://doi.org/10.7554/eLife.39163.044
Author response https://doi.org/10.7554/eLife.39163.045

## Additional files

### Supplementary files

• Supplementary file 1. Data collection and refinement statistics. Data collection and refinement statistics for the presented cryoEM maps and structural models. VPP = volta phase plate.
DOI: https://doi.org/10.7554/eLife.39163.023

• Transparent reporting form
DOI: https://doi.org/10.7554/eLife.39163.024

### Data availability

The atomic coordinates for the Rea1 AAA+ ring and the Rea1 linker in the ADP state have been deposited with PDB IDs 6HYP and 6HYD, respectively. The accession codes for the Rea1 and Rea1_ΔAAA2L-H2α models in the AMPPNP state are 6I26 and 6I27, respectively. The accession codes for the cryoEM maps of the Rea1 AAA+ ring and the Rea1 linker in the ADP state are EMD-0309 and EMD-0308, respectively. The accession code for the unsharpened cryoEM map of the Rea1 AAA+ ring in the ADP state is EMD-0330. The cryoEM maps of the Rea1 and Rea1_ΔAAA2L-H2α AMPPNP states have the EMD accession codes EMD-0328 and EMD-0329, respectively.

The following datasets were generated:

| Author(s) | Year | Dataset title | Dataset URL | Database and Identifier |
|---|---|---|---|---|
| Schmidt H, Sosnowski P, Busselez J, Fagiewicz R | 2018 | Rea1 AAA+ ring in the ADP state | http://www.rcsb.org/structure/6HYP | RCSB Protein Data Bank, 6HYP |
| Schmidt H, Sosnowski P, Busselez J, Fagiewicz R | 2018 | Rea1 linker in the ADP state | http://www.rcsb.org/structure/6HYD | RCSB Protein Data Bank, 6HYD |
| Schmidt H, Sosnowski P, Busselez J, Fagiewicz R | 2018 | Rea1 AMPPNP structure | http://www.rcsb.org/structure/6I26 | RCSB Protein Data Bank, 6I26 |
| Schmidt H, Sosnowski P, Busselez J, Fagiewicz R | 2018 | Rea1_ΔAAA2L-H2α AMPPNP structure | http://www.rcsb.org/structure/6I27 | RCSB Protein Data Bank, 6I27 |
| Schmidt H, Sosnowski P, Busselez J, Fagiewicz R | 2018 | Sharpened map of the Rea1 AAA+ ring ADP state | http://www.ebi.ac.uk/pdbe/entry/emdb/EMD-0309 | Electron Microscopy Data Bank, EMD-0309 |
| Schmidt H, Sosnowski P, Busselez J, Fagiewicz R | 2018 | Sharpened map of the Rea1 linker ADP state | http://www.ebi.ac.uk/pdbe/entry/emdb/EMD-0308 | Electron Microscopy Data Bank, EMD-0308 |
| Schmidt H, Sosnowski P, Busselez J, Fagiewicz R | 2018 | Unsharpened map of the Rea1 AAA+ ring ADP state | http://www.ebi.ac.uk/pdbe/entry/emdb/EMD-0330 | Electron Microscopy Data Bank, EMD-0330 |
| Schmidt H, Sosnowski P, Busselez J, Fagiewicz R | 2018 | Sharpened map of the Rea1 AMPPNP state | http://www.ebi.ac.uk/pdbe/entry/emdb/EMD-0328 | Electron Microscopy Data Bank, EMD-0328 |
| Schmidt H, Sosnowski P, Busselez J, Fagiewicz R | 2018 | Sharpened map of the Rea1_ΔAAA2L-H2α AMPPNP state | http://www.ebi.ac.uk/pdbe/entry/emdb/EMD-0329 | Electron Microscopy Data Bank, EMD-0329 |

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
