## [Decision Letter]

Thank you for submitting your article "The CryoEM Structure of the Ribosome Maturation Factor Rea1" for consideration by *eLife*. Your article has been reviewed by three peer reviewers, one of whom is a member of our Board of Reviewing Editors, and the evaluation has been overseen by Andrea Musacchio as the Senior Editor. The reviewers have opted to remain anonymous.

The reviewers have discussed the reviews with one another and the Reviewing Editor has drafted this decision to help you prepare a revised submission.

Summary:

The study presented by Sosnowski and Urnavicius et al. reports the first atomistic model of the Rae1 ribosome assembly factor. Rae1 belongs to the AAA family of AAA ATPase and participates in the remodeling of the large ribosomal subunit by removing several smaller associated factors during 60S biogenesis. While the importance of Rae1 in this pathway is clearly established, the molecular mechanism of these remodeling reactions is still poorly understood in large part because of a lack of structural information on the Rae1 AAA ATPase. This study presents the cryo-EM structure of *S. cerevisiae* Rae1, which reveals several novel structural features for this AAA ATPase and attempts to present a plausible mechanism for its removal of effectors from the ribosome through the MIDAS domain that is disordered and not visible in this structure. This is a straightforward structure paper that will no doubt catalyze interesting mechanistic studies and it should be published.

Essential revisions:

The reviewers agreed that there were four areas that required strengthening before the work can be published: (1) Validation of the structure; (2) A deeper discussion of the structure; (3) A better and more detailed discussion of their mechanistic model; and (4) Some testing of that model. Details on these four areas follow:

1) Validation of the structure

- The manuscript is surprisingly lacking in any validation of the structure. None of the standard metrics in the field have been reported. The authors should provide the following: FSC (gold standard) curves for all refinements; FSC (model-to-map) for their final models; maps showing local resolution; locally-filtered maps (is the map in Figure 1—figure supplement 1 locally-filtered?); table with MolProbity scores and Ramachandran statistics. (Although the authors did submit a PDB validation report, these statistics need to be in the manuscript itself.) Since the final model is a composite of structures that were refined separately, these statistics should be reported for each structure. Similarly, a table (or tables) with data collection and refinement statistics should be included as a supplementary figure. It is not sufficient to include this information in the Materials and methods.

- In Materials and methods it is stated that the authors used a total electron dose of 6.6 electrons/A^2^ ? This seems low. Can the authors check or comment?

2) Deeper discussion of the structure

- In terms of interpretation of the structure, the authors find that AAA6-AAA1 and AAA1-AAA2 interfaces are in the closed conformation. Most often, AAA+ motors only adopt a closed configuration in the ATP state, as the arg finger from the adjacent subunit contacts the ATP γ-phosphate. In the absence of the γ-phosphate, We would expect that the interfaces would resemble the post-hydrolysis 'open' state. It is interesting that these are the two sites that are expected to be incompetent for ATP hydrolysis. We feel that the authors should comment on this unexpected finding.

- Given the parallels between Rae1 and dynein, the authors should include a structural comparison between these proteins in their Discussion.

- The significance of the nucleotide state of the structure is unclear. Why was the structure solved in the presence of ADP? Since AAA1 and AAA6 lack the catalytic residue in the Walker B motif, isn't the expectation that in vivo these two subunits are always bound to ATP, not ADP (as in the reconstruction)? Can the authors comment on this? Do they have any information on an ATP bound state? What was the reason for assembling Rea1 complex in excess ADP and not ATP? What functional state do the authors think their complex represents?

3) Better and more detailed discussion of their mechanistic model

- The structure should reveal insights into the mechanism of Rea1 activity. Unfortunately, the mechanism proposed in the current manuscript is too opaque. Figure 4 should summarize the findings and indicate the authors' proposal for how Rea1 modifies the pre-60S ribosome structure. The figure does have a small conformational change that the authors propose is important for regulating ATPase activity. While this regulation is of moderate interest, this was not convincingly shown. We would hope that the structure would lead to more insights than simply 'AAA+ communicates with the linker segment' which was not a new insight. In fact, the figure shows no conformational change in this segment despite the authors spending much time in Figure 3—figure supplement 1 discussing how the ATPase activity drives conformational changes in the linker. The section describing these conformational changes was confusing and not very informative. The authors allude to a conformational change at the AAA+/stem connection, but don't describe their thoughts cogently. Alternatively, it seemed as if the authors were proposing that the Top domains may move in a ball-in-socket fashion around the middle/stem regions. Furthermore, Figure 4 makes it appear that there is no conformational change in the linker segment, which was confusing considering the discussion of conformational change. What exactly are the authors proposing for the Ytm1/Rsa4 removal mechanism?

- Given the length of the flexible D/E rich region (over 600 residues), what is the mechanism of force transmission through the MIDAS domain, which is the one that binds to the substrate? Wouldn't the length and low complexity of the D/E region allow the MIDAS domain to freely interact with its substrate anyway without needing the motion of the linker?

4) Testing of the model

- The authors should present some data from experiments testing their mechanistic model. The reviewers agreed that it should be possible to make structural predictions about the transmission of conformational changes from the motor to the linker. These could be either functional assays, or some type of EM that shows that conformational changes the model predicts do take place.

---

## [Author Response]

Essential revisions:The reviewers agreed that there were four areas that required strengthening before the work can be published: (1) Validation of the structure; (2) A deeper discussion of the structure; (3) A better and more detailed discussion of their mechanistic model; and (4) Some testing of that model. Details on these four areas follow:1) Validation of the structure- The manuscript is surprisingly lacking in any validation of the structure. None of the standard metrics in the field have been reported. The authors should provide the following: FSC (gold standard) curves for all refinements; FSC (model-to-map) for their final models; maps showing local resolution; locally-filtered maps (is the map in Figure 1—figure supplement 1 locally-filtered?); table with MolProbity scores and Ramachandran statistics. (Although the authors did submit a PDB validation report, these statistics need to be in the manuscript itself.) Since the final model is a composite of structures that were refined separately, these statistics should be reported for each structure. Similarly, a table (or tables) with data collection and refinement statistics should be included as a supplementary figure. It is not sufficient to include this information in the Materials and methods.

This has been done. We now provide gold standard FSC curves for the refinements, model-to-map FSC’s of the structural models as well as local resolution maps for the Rea1 ADP AAA+ ring and linker maps (Figure 1—figure supplement 1), the Rea1 AMPPNP map (Figure 4—figure supplement 1) and the Rea1_ΔAAA2L-H2α AMPPNP map (Figure 4—figure supplement 5). MolProbity scores and Ramachandran statistics for the AAA+ ring and linker models are provided in Supplementary file 1. We don’t report these values for the Rea1 AMPPNP and Rea1_ΔAAA2L-H2α AMPPNP structures, because they were created via rigid-body docking of the Rea1 ADP structures. CryoEM data collection statistics are also provided in Supplementary file 1.

- In Materials and methods it is stated that the authors used a total electron dose of 6.6 electrons/A^2^ ? This seems low. Can the authors check or comment?

This has been corrected. We accidentally reported the dose per second and not the total dose. The total dose varied between 45 and 50 electrons/A^2^ for the three data sets reported here. We have corrected this in the Materials and methods section and also report these values in Supplementary file 1.

2) Deeper discussion of the structure- In terms of interpretation of the structure, the authors find that AAA6-AAA1 and AAA1-AAA2 interfaces are in the closed conformation. Most often, AAA+ motors only adopt a closed configuration in the ATP state, as the arg finger from the adjacent subunit contacts the ATP γ-phosphate. In the absence of the γ-phosphate, We would expect that the interfaces would resemble the post-hydrolysis 'open' state. It is interesting that these are the two sites that are expected to be incompetent for ATP hydrolysis. We feel that the authors should comment on this unexpected finding.

This has been done. We now discuss the degree of the closure of the Rea1 AAA1 and AAA6 nucleotide binding sites in the section”Structure of the Rea1 AAA+ ring”. We compare the Rea1 AAA1 and AAA6 sites to closed and more open AAA sites in the Rea1 related dynein motor.

- Given the parallels between Rae1 and dynein, the authors should include a structural comparison between these proteins in their Discussion.

This has been done. We have now included a section entitled “Comparison of the Rea1 and dynein AAA+ rings” analysing the similarities between both proteins. In this paragraph we point out that the geometry of the Rea1 AAA+ ring closely resembles the dynein AAA+ ring in the ADP state. The geometry of the rings can be roughly described as two halves of three domains each. In one half, the AAA+ domains are more tightly associated. In the other half they are more loosely packed. In both cases the loosely packed half of the AAA+ ring features a central AAA+ domain that is rotated with respect to the other domains.

- The significance of the nucleotide state of the structure is unclear. Why was the structure solved in the presence of ADP? Since AAA1 and AAA6 lack the catalytic residue in the Walker B motif, isn't the expectation that in vivo these two subunits are always bound to ATP, not ADP (as in the reconstruction)? Can the authors comment on this? Do they have any information on an ATP bound state? What was the reason for assembling Rea1 complex in excess ADP and not ATP? What functional state do the authors think their complex represents?

We initially focused on the Rea1 ADP state because one of the first high-resolution crystal structures of the Rea1 related dynein motor was obtained in the presence of ADP (Kon et al., 2012). We hypothesised that ADP might also have a stabilizing effect on Rea1 so that a potentially higher resolution might be achieved compared to other nucleotide states. We did not choose ATP because we suspected that ongoing ATP hydrolysis in our sample would introduce additional conformational heterogeneity that would lower our chances of obtaining a high-resolution cryoEM reconstruction of Rea1.

Concerning the nucleotide states of AAA1 and AAA6, we tried to fit ATP instead of ADP. However, there was not enough space for the γ-phosphate. The cryoEM structure was also obtained in the presence of 3 mM ADP which makes it likely that the observed nucleotide is ADP.

We now provide additional cryoEM structures of the AMPPNP state for wildtype Rea1 and a mutant where the AAA2L-H2α insert (the central “plug” of the ring) has been deleted. In line with our original proposal of AAA2L-H2α as auto-inhibitory regulator, these structures now clearly demonstrate that important nucleotide dependent conformational rearrangements of the ring only take place when AAA2L-H2α is not present at its center. The Rea1 ADP structure in the original manuscript as well as the additional Rea1 AMPPNP in the revised version represent the identical, auto-inhibited Rea1 state.

3) Better and more detailed discussion of their mechanistic model- The structure should reveal insights into the mechanism of Rea1 activity. Unfortunately, the mechanism proposed in the current manuscript is too opaque. Figure 4 should summarize the findings and indicate the authors' proposal for how Rea1 modifies the pre-60S ribosome structure. The figure does have a small conformational change that the authors propose is important for regulating ATPase activity. While this regulation is of moderate interest, this was not convincingly shown. We would hope that the structure would lead to more insights than simply 'AAA+ communicates with the linker segment' which was not a new insight. In fact, the figure shows no conformational change in this segment despite the authors spending much time in Figure 3—figure supplement 1 discussing how the ATPase activity drives conformational changes in the linker. The section describing these conformational changes was confusing and not very informative. The authors allude to a conformational change at the AAA+/stem connection, but don't describe their thoughts cogently. Alternatively, it seemed as if the authors were proposing that the Top domains may move in a ball-in-socket fashion around the middle/stem regions. Furthermore, Figure 4 makes it appear that there is no conformational change in the linker segment, which was confusing considering the discussion of conformational change. What exactly are the authors proposing for the Ytm1/Rsa4 removal mechanism?

In order to better distinguish between the structural description of Rea1 and our ideas about the Rea1 mechanism, we have split “Results and Discussion” of the original manuscript in two separate “Results” and “Discussion” sections in the revised version. In our new “The AAA+ ring – linker interface and the structure of the Rea1 linker” paragraph of the Results section we now simply describe our findings about the structural elements that make up the AAA+ ring – linker interface and point out the architecture of the linker.

We have removed the speculative parts about movements and interactions of AAA+ domains AAA6, AAA1, AAA5 as well as AAA2-AAA4 and how these might introduce linker remodelling from the “Discussion” section of the revised version. We now simply remind the reader that AAA6 is the only AAA+ domain that directly contacts the linker via interactions with the linker stem and the helix extension of the middle domain, which are completely new insights that were not obvious from previous work. We go on to briefly point out that a potential movement of AAA6 during ATP hydrolysis in the ring could be transferred to the linker middle domain via the interaction between AAA6 and the helix extension of the middle domain. A conformational shift of the middle domain might trigger a large scale remodelling event given the importance of the middle domain for the architecture of the linker top. We also clearly state that we did not observe such an event despite testing various nucleotide states of Rea1 and the Rea1_ΔAAA2L-H2α mutant and discuss potential reasons for this unexpected finding.

We also added the additional aspect of the AAA+ ring MIDAS domain binding site to our Discussion. We provide strong evidence that this binding site would place the MIDAS domain in direct contact with the Rsa4 substrate when Rea1 is bound to pre60S particles and that the formation of the MIDAS domain binding site is controlled by the auto-inhibitory AAA2L-H2α insert. We believe that these findings provide important additional insights into the Rea1 mechanism and that they have significantly strengthened our manuscript.

- Given the length of the flexible D/E rich region (over 600 residues), what is the mechanism of force transmission through the MIDAS domain, which is the one that binds to the substrate? Wouldn't the length and low complexity of the D/E region allow the MIDAS domain to freely interact with its substrate anyway without needing the motion of the linker?

We discuss this issue now in the “Discussion” section of our revised manuscript. Since the MIDAS domains is fixed at the Rea1 AAA+ ring when it binds to its Rsa4 substrate, a linker remodelling event might simply stretch the flexible D/E region to transmit the force for Rsa4 removal to the MIDAS domain.

4) Testing of the model- The authors should present some data from experiments testing their mechanistic model. The reviewers agreed that it should be possible to make structural predictions about the transmission of conformational changes from the motor to the linker. These could be either functional assays, or some type of EM that shows that conformational changes the model predicts do take place.

We decided to follow the structural approach suggested by the reviewers since structural information is ultimately needed to understand the molecular events causing Rea1 linker remodelling. We determined cryoEM structures of the Rea1 AMPPNP state and negative stain electron microscopy structures of the Rea1 APO and ATP states. However, we were not able to observe linker remodelling. We hypothesised that the auto-inhibitory AAA2L-H2α insert might have prevented linker remodelling and continued to determine negative

stain electron microscopy structures of the Rea1_ΔAAA2L-H2α APO, AMPPNP, ATP and ADP states. Furthermore, we also determined a cryoEM structure of the Rea1_ΔAAA2L-H2α AMPPNP state. In all these cases we did not observe Rea1 linker remodelling.

These results strongly suggest that linker remodelling on isolated Rea1 molecules cannot be induced by simply adding different nucleotides. Rea1 behaves clearly differently from the related dynein motor, where linker remodelling correlates with defined nucleotide states. In the case of Rea1, additional factors seem to be needed to induce such an event. One such potential factor could be the pre60S particle that might provide important interactions for Rea1 linker remodelling.

Since the conditions under which linker remodelling occurs could not be identified, it is at the moment impossible to evaluate the effect of mutations. Even if we would observe nucleotide dependent linker remodelling in a mutant, we would not be able to distinguish if such an event is physiological or an artefact.